# Mapping the Melatonin Suppression, Star Light and Induced Photosynthesis Indices with the LANcube

**Martin Aubé [1,2,3,\*], Charles Marseille [2], Amar Farkouh [1], Adam Dufour [1], Alexandre Simoneau [2], Jaime Zamorano [4,5], Johanne Roby [1] and Carlos Tapia [4]**

1   Cégep de Sherbrooke, Département de Géomatique Appliquée, Université de Sherbrooke, Sherbrooke, QC J1E 4K1, Canada; amar.farkouh@umontreal.ca (A.F.); adam.dufour@usherbrooke.ca (A.D.); johanne.roby@cegepsherbrooke.qc.ca (J.R.)
2   Département de Géomatique Appliquée, Université de Sherbrooke, Sherbrooke, QC J1K 2R1, Canada; charles.marseille@usherbrooke.ca (C.M.); alexandre.simoneau@usherbrooke.ca (A.S.)
3   Physics Department, Bishop's University, Sherbrooke, QC J1M 1Z7, Canada
4   Department Física de la Tierra y Astrofísica, Universidad Complutense de Madrid, 28040 Madrid, Spain; jzamorano@fis.ucm.es (J.Z.); carloseugeniotapia@ucm.es (C.T.)
5   Instituto de Física de Partículas y del Cosmos (IPARCOS), Universidad Complutense de Madrid, 28040 Madrid, Spain
\*   Correspondence: martin.aube@cegepsherbrooke.qc.ca; Tel.: +1-819-564-6350 (ext. 4146)

**Abstract:** Increased exposure to artificial light at night can affect human health including disruption of melatonin production and circadian rhythms which can extend to increased risks of hormonal cancers and other serious diseases. In addition, multiple negative impacts on fauna and flora are well documented, and it is a matter of fact that artificial light at night is a nuisance for ground-based astronomy. These impacts are frequently linked to the colour of the light or more specifically to its spectral content. Artificial light at night is often mapped by using spaceborne sensors, but most of them are panchromatic and thus insensitive to the colour. In this paper, we suggest a method that allows high-resolution mapping of the artificial light at night by using ground-based measurements with the LANcube system. The newly developed device separates the light detected in four bands (Red, Green, Blue and Clear) and provides this information for six faces of a cube. We found relationships between the LANcube's colour ratios and (1) the Melatonin Suppression Index, (2) the StarLight Index and (3) the Induced Photosynthesis Index. We show how such relationships combined with data acquisition from a LANcube positioned on the top of a car can be used to produce spectral indices maps of a whole city in a few hours.

**Keywords:** artificial light at night; intrusive light; direct light pollution; radiometry; multispectral; multiangular; Melatonin Suppression Index; Star Light Index; spectroscopy; measurement; synthetic photometry

## 1. Introduction

For several decades, the increase of artificial light at night (ALAN) has greatly altered the nocturnal integrity. ALAN contributes to the disruption of daily, lunar and seasonal cycles of natural light [1] and has significant consequences on fauna [1,2], flora [3], the starry sky [4] and human health [5].

It is well known that day/night light cycles regulate circadian rhythms and synchronise the circadian clock in humans [6] and other animals. Light has also been shown to impact several physiological functions such as melatonin suppression [7]. Indeed, melatonin is a hormone produced by the pineal gland in relation to the light detected by a non-visual retina photoreceptor, the melanopsin.

This photoreceptor has a maximum spectral sensitivity at blue wavelengths [8,9]. There is strong evidence that many biological effects of artificial light at night (ALAN) are directly related to its spectral content. For example, some studies have shown that exposure to blue-enriched light may increase the risk of hormone-dependent cancers [10–12]. One accurate way to detect the impact of ALAN on circadian systems is to monitor the level of melatonin in biofluids [13], but such a procedure can be complex to perform on large samples and/or large areas.

In 2013, Aubé et al. [14] proposed the Melatonin Suppression Index (MSI) as a new metric to evaluate the potential effect of the spectral distribution of the light on the melatonin suppression mechanism. The MSI only requires the knowledge of the ambient spectral content of ALAN to be determined. This index was designed to disentangle the effect associated with the spectral content from the one related to the absolute visual light level. The MSI typically ranges on a scale from 0 to ≈1, 0 being a spectrum with no impact on melatonin suppression and 1 being the impact of the sunlight spectrum. The sun-like CIE standard illuminant D65 [15] was taken as a reference in the calculation of this index because the sun is the main natural source of light under which the human body adapted over time. It contains the whole visible part of the light spectrum. The MSI uses a fit of the Melatonin Suppression Action Spectrum (MSAS) data acquired by Thapan et al. [16] and Brainard et al. [9] as a weighting function. The MSI is mostly affected by blue-enriched light due to the high sensitivity of the melanopsin to this spectral range [17]. Lamps can have a MSI higher than 1 even if it is not common in typical lighting technologies. Garcia-Saenz et al. [10,11] used the MSI as a proxy for the blue light content effect on breast, prostate and colorectal cancers in Madrid and Barcelona. They found correlations between the MSI from street lights nearby homes of participants to the MCC-Spain epidemiological study for the three cancers evaluated.

In a similar way to the MSI, the Star Light Index (SLI; [14]) can be used to monitor the impact of the spectrum of ALAN on stellar observations. The threat to astronomical observation capacities is caused by the scattering of ALAN by molecules and aerosols in the atmosphere. The light travelling upward can then be redirected to an observer and hence compete with the low light levels coming from astronomical objects. The SLI also typically spans from 0 to ≈1, 0 being a colour that does not compete with stars visibility and 1 being a colour that affects star visibility like a D65 spectrum would do. The SLI uses the scotopic sensitivity as a weighting function. Blue-enriched light has also revealed to give higher SLI values, because that the SLI is more sensitive in the blue region.

Finally, the Induced Photosynthesis Index (IPI; [14]) can be used to monitor the impacts of the spectrum of artificial lights on plants. As for the MSI and the SLI, the IPI typically spans from 0 to ≈1, 0 being a colour that does not stimulate the photosynthesis process and 1 being a colour that affects photosynthesis like a D65 spectrum would do. The IPI uses the Photosynthesis Action Spectrum (PAS; DIN5031–10 [18]) as a weighting function.

The calculation of the MSI, SLI and IPI for a given light radiant flux depends on the shape of its spectral power distribution. This calculation requires the use of a spectrometer to sample the lamp spectrum. However, spectral measurements are not the easiest to obtain, especially if many of them are required. It has been shown that three-band Red, Green, Blue (RGB) photometric data can be used to determine spectral indices or photoreceptor stimuli from values recorded by a Digital single-lens reflex camera (DSLR) [19,20]. This method is much faster and easier to perform.

In this paper, we developed a similar method of calculating the MSI, SLI and IPI out of four-band RGB and Clear (RGBC) photometric data detected by the newly developed LANcube system. This method relies on a cross-calibration between RGBC values of the LANcube obtained using synthetic photometry method applied on a large spectral database along with the MSI, SLI and IPI determination by the integration of the same spectra.

Very few methods are available to map the measured ALAN-related parameters over large territories. A number of instruments may help to perform such remote sensing tasks, with each instrument having its specific capacities and limitations. Hänel et al. [21] provided a good review of the most common instruments available. Basically they can be separated in two classes: imaging

and non-imaging devices. For each class, we can distinguish between spectrum sensitive devices and panchromatic devices. For the sake of the present work, where we want to determine spectral indices, panchromatic devices cannot be of any help. It is clear that colour imaging systems installed on airborne or spaceborne platforms are well suited for such mapping. This can be performed using stratospheric balloons experiments [22,23], astronauts photography taken from the International Space Station [20] of from unmanned aerial observing platforms [24,25]. However, there are very few stratospheric balloon flights being performed for that purpose and astronauts are more likely to shot at large emblematic cities than any other site. Unmanned aerial platforms are still sparsely available but they offer increasing potential as they become more and more readily common. In this paper, we show how the LANcube system installed on top of a moving vehicle can be used as an alternative to spaceborne and aerial imaging techniques.

## 2. Materials and Methods

In this section, we explain how the raw RGBC values of the LANcube can be converted into the MSI, SLI and IPI after determining relationships between colour–colour ratios and each index.

### 2.1. The LANcube System

The LANcube, also known as LAN$^3$, is a new device developed by Pr. Aubé's research group. It is intended to sample the multispectral and multidirectional properties of the Direct Artificial Light At Night (DALAN) into any environment (indoor or outdoor). It was designed to evaluate the impact of DALAN on human health and ecosystems.

The LAN$^3$ is a cube-shaped device having a sensor with four visible spectral bands (red, green, blue and clear) on each of its six faces. The spectral sensitivity of each band is shown in Figure 1. The sensor used is the TCS 34725 colour sensor manufactured by Texas Advanced Optoelectronic Solutions Inc., Plano, Texas, United States (TAOS). A watch glass protects the sensor electronics. The spectral response of each band, including the effect of the spectral transmittance of the watch glass, was determined at the LICA optical laboratory of the Universidad Complutense de Madrid. We used a monochromator, a calibrated photodiode and an integration sphere. The sensor has a Field of View that is very close to a cosine function. Aside from the light sensors, the LAN$^3$ contains a real-time clock, a micro-SD card to store the data, a GPS module and a temperature sensor. The raw Analog to Digital Units (ADU) recorded in each band, the integration time and the gain, are stored on the SD card along with the date, time, GPS coordinates, and temperature. The first version of the LAN$^3$ (LAN$^3$v1) uses the Arduino open source microcontroller. All the required documentation to learn how to build the device is available online [26].

A new version of the device (LAN$^3$v2) will be released soon. It will have shorter acquisition time. That version will allow an access to the data from a mobile device. The Arduino microcontroller is replaced by a Raspberry Pi computer. In addition, the LAN$^3$v2 has an Uninterruptible Power Supply (UPS) to protect the system against power failures. Pictures of LAN$^3$v1 and LAN$^3$v2 are shown in Figure 2. A quick comparison of the the two versions is available in Table 1.

The minimum light level detected is of the order of 0.015 lux (Signal to Noise Ratio $\geq$ 3). The LAN$^3$ is equipped with an automated gain and integration time adjustment algorithm allowing the setup of optimal values that maximise the photometric resolution of each light sensor while keeping the integration time as low as possible. This procedure aim to maximise the data acquisition rate.

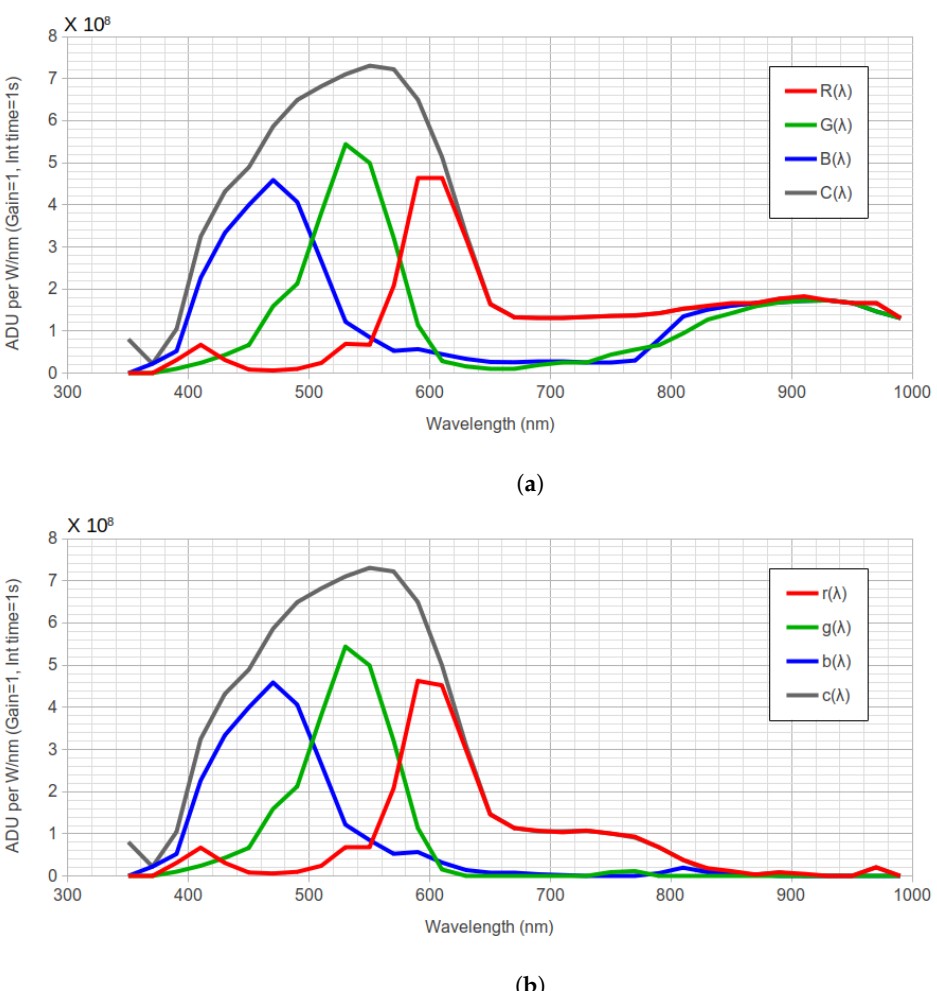

(**a**)

(**b**)

**Figure 1.** Spectral response of the LAN[3] as determined at the LICA laboratory at Universidad Complutense de Madrid. The calibration curves are for a gain of 1 and an integration time of 1 s. In panel (**a**) we show the original spectral responses $R(\lambda)$, $G(\lambda)$, $B(\lambda)$, and $C(\lambda)$ (Red, Green, Blue and Clear bands). In panel (**b**) we show $r(\lambda)$, $g(\lambda)$, $b(\lambda)$ and $c(\lambda)$, which are the band responses after removal of an estimate of the Infrared contribution (see Equations (1) and (2)).

**Table 1.** Features comparison of LAN[3] versions.

| Feature | v1 | v2 |
|---|---|---|
| Sensors | TCS 34725 | TCS 34725 |
| Number of sensors | 6 | 5 |
| Minimum acquisition time | 7 s | 1 s |
| Maximum acquisition time | 11 s | 2 s |
| Processing unit | Arduino atMega 2560 | Raspberry Pi 4b computer |
| Access to data | Removal of the microSD card | Download from wifi |
| System state indicator | RGB LED | RGB LED + wifi web server |
| Environment sensor | DTH22 (temp. and Hum.) | Onboard Raspberry pi temp. sensor |
| Power source | 5 V—USB/A | 5 V—USB/C |
| Power protection | Nothing | Integrated UPS |
| Positioning | GPS | GPS |
| Real time clock | DS 3231 | DS 3231 |

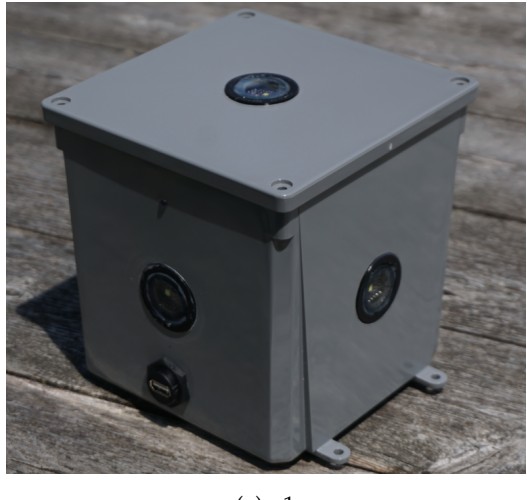

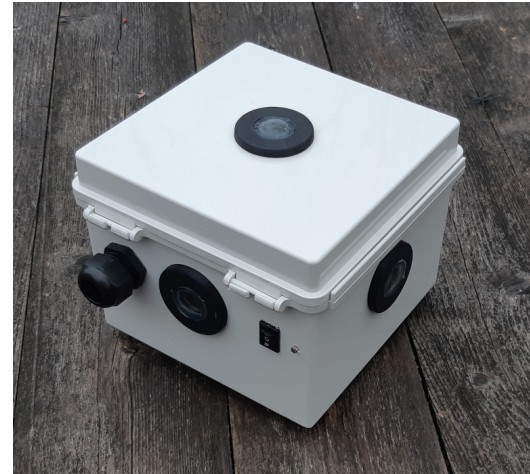

|(**a**) v1|(**b**) v2|

**Figure 2.** The LAN$^3$ devices.

### 2.2. Synthetic Photometry

With the spectral responses of the sensor/glass window known, synthetic photometry can be used to simulate the signal that must be detected by the LAN$^3$ for any given spectrum. This is done by taking the weighted integral of the spectra over each of the sensors bands with the known spectral response of each band (see Figure 1a). To convert the RGBC response to rgbc, we removed an estimate of the infrared signal (IR) by using RGBC signals (see Figure 1b). The infrared spectral response can be computed using Equation (1).

$$IR(\lambda) \approx \frac{R(\lambda) + G(\lambda) + B(\lambda) - C(\lambda)}{2} \tag{1}$$

The resultant spectral sensitivity in the infrared ($IR(\lambda)$) is presented in Figure 3.

Therefore, $r(\lambda), g(\lambda), b(\lambda)$ and $c(\lambda)$ can be obtained with Equation (2).

$$r(\lambda) \approx R(\lambda) - IR(\lambda) \quad g(\lambda) \approx G(\lambda) - IR(\lambda) \quad b(\lambda) \approx B(\lambda) - IR(\lambda) \quad c(\lambda) \approx C(\lambda) - IR(\lambda) \tag{2}$$

Finally, the readings excluding the infrared are given by Equation (3),

$$r = \int J(\lambda)\, r(\lambda)\, d\lambda \qquad g = \int J(\lambda)\, g(\lambda)\, d\lambda \qquad b = \int J(\lambda)\, b(\lambda)\, d\lambda \qquad c = \int J(\lambda)\, c(\lambda)\, d\lambda \tag{3}$$

where $r$, $g$, $b$ and $c$ are the simulated ADU values of the LAN$^3$ sensor excluding the infrared, when looking to a given light radiant flux. $J(\lambda)$ is the spectrum of the given light radiant flux and $r(\lambda), g(\lambda)$ and $b(\lambda)$ are, respectively, the spectral response of the Red, Green and Blue bands of the LAN$^3$ after removal of the infrared response.

In order to establish a relationship between LAN$^3$ measurements and MSI/SLI/IPI values, 314 different spectra were measured with a Stellarnet's Black comet spectrometer. The spectrometer operates in the spectral range of 280 to 900 nm with a resolution of 1 nm. The sample set contains 172 street lights, indoor lamps and other types of conventional lighting commonly available. These spectra are coming from the Lamp Spectral Power Distribution Database (LSPDD; Roby et al. [27]). The remaining 142 spectra were acquired in situ. This allowed us to have mixed spectra combining multiple sources and environments. The wide variety of spectra is a key to establish a robust relationship over the colour ratios space defined by the $\left[\frac{b}{g}\right]$ and $\left[\frac{r}{g}\right]$.

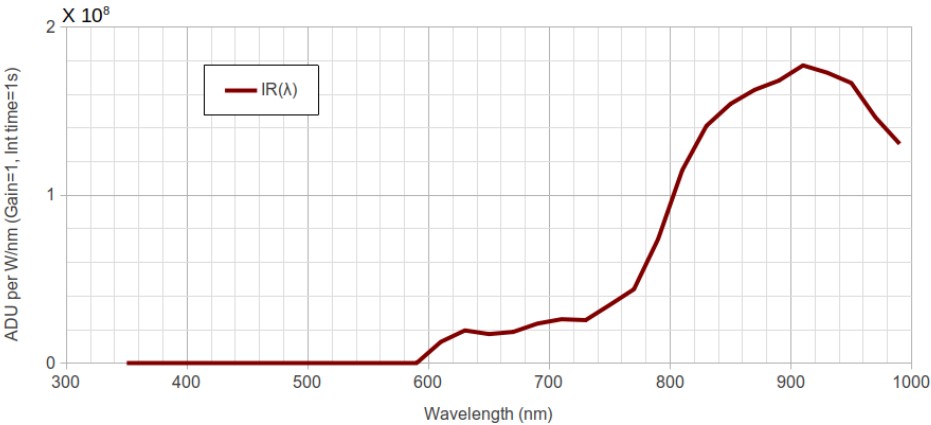

**Figure 3.** Estimated infrared spectral response.

### 2.3. Calculation of the Indices from Spectral Data

The calculation of the MSI, SLI and IPI is done using the method described in Aubé et al. [14] over the LSPDD and in situ spectra. The spectral responses used to calculate the indices come from Aubé et al. [14] for MSAS, Vos [28] for Scotopic ($V'(\lambda)$), DIN5031–10 [18] for PAS and Wyszecki and Stiles [29] for Photopic ($V(\lambda)$). These spectral responses can be downloaded from the LSPDD. The CIE standard illuminant D65 spectrum, also used in the indices calculations, was taken from the LSPDD.

### 2.4. Fitting the Indices 3D Surface in the $\left[\frac{b}{g}\right] - \left[\frac{r}{g}\right]$ Space

Once the MSI, SLI and IPI values are calculated with the spectra, they are compared with the LAN³ colour ratios $\left[\frac{b}{g}\right]$ and $\left[\frac{r}{g}\right]$ of each spectrum as determined by the synthetic photometry method explained in Section 2.2. This creates three 3D spaces, one for MSI (Figure 4), one for SLI and one for IPI, with each one having 314 data points. For each 3D space, a 3D surface is fitted to the data and the Equation (4) of this surface becomes the transformation equation between the LAN³ colour ratios and the respective index value. The equation is the same for each index but the parameters $\alpha$ to $\zeta$ differ. The purpose of the division by the $g$ value is to remove the effect of absolute luminous intensity on the radiometric measurements of the instrument. We used a Python code to fit a 2nd order polynomial 3D surface into the data for each index (see Equation (4)).

$$Index = \alpha + \beta \left[\frac{b}{g}\right] + \gamma \left[\frac{r}{g}\right] + \delta \left[\frac{b}{g}\right]\left[\frac{r}{g}\right] + \epsilon \left[\frac{b}{g}\right]^2 + \zeta \left[\frac{r}{g}\right]^2 \tag{4}$$

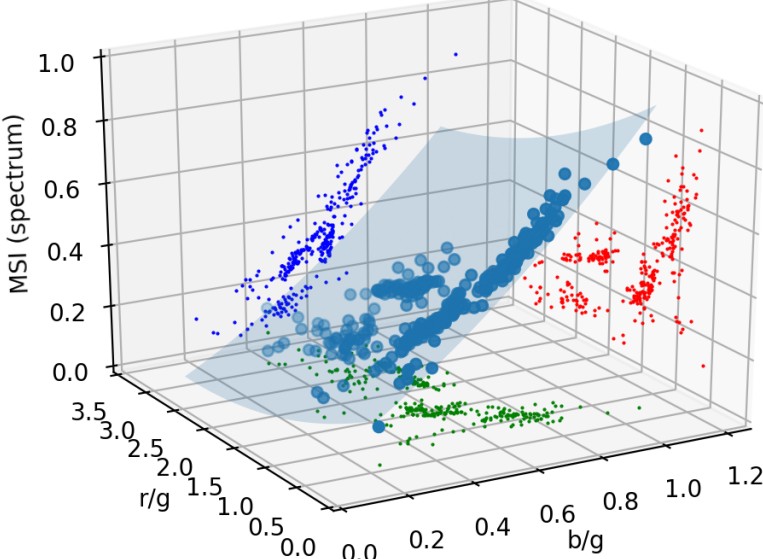

**Figure 4.** Fit of a second-order polynomial surface for the Melatonin Suppression Index. The light blue circles are the 314 spectra. The $r$, $g$ and $b$ values are determined from synthetic photometry using the $r(\lambda)$, $g(\lambda)$ and $b(\lambda)$ spectral responses shown in Figure 1b. The dots are the projection of the point on each orthogonal plane (red $= \left[\frac{r}{g}\right] -$ MSI; green $= \left[\frac{b}{g}\right] - \left[\frac{r}{g}\right]$; dark blue $= \left[\frac{b}{g}\right] -$ MSI).

## 3. Results

The method described above was applied to a large variety of spectral power distributions and indices. The results for MSI is presented in Figure 4. In that figure, the light blue circles are the 314 spectra. The light blue transparent surface is the result of a fit of a 3D 2nd order polynomial surface to the whole dataset. The fitted parameters of Equation (4) for each index are given in Table 2.

**Table 2.** Fitted coefficients of Equation (4) for the MSI, SLI and IPI.

| Index | $\alpha$ | $\beta$ | $\gamma$ | $\delta$ | $\epsilon$ | $\zeta$ |
|---|---|---|---|---|---|---|
| MSI | 0.0769 | 0.6023 | −0.1736 | −0.0489 | 0.3098 | 0.0257 |
| SLI | 0.6624 | −0.4308 | −0.2891 | 0.2913 | 0.6801 | 0.0015 |
| IPI | −0.4118 | −0.3824 | −0.0955 | 0.7048 | 0.3305 | −0.0463 |

For the three indices, the most important colour ratio is $\left[\frac{b}{g}\right]$. This reflects in the higher values of $\beta$ and $\epsilon$. Nevertheless, $\left[\frac{r}{g}\right]$ cannot be neglected at all. In the search for the optimal equation, we tried lower order polynomial functions but the results were not satisfying. In order to evaluate the intrinsic errors associated with the use of the fitted equations, we used them to calculate the three indices and compared the resultant indices to the accurate indices calculations obtained with the spectra. This is shown in Figures 5–7. The figures show a good correlation between the two ways of determining the indices. In these figures, the solid line is the 1:1 relation and the right panel of each figure shows the residuals. For the MSI, the standard deviation of the residuals is 0.024 while it is 0.056 for the SLI. The correlation is slightly lower for the IPI with a standard deviation of 0.107. The standard deviations translate in the margin of error (95 percent confidence interval) of ±0.05 for MSI, ±0.1 for SLI, and ±0.2 for IPI. The complete dataset used in the fit is available on the project github repository [30].

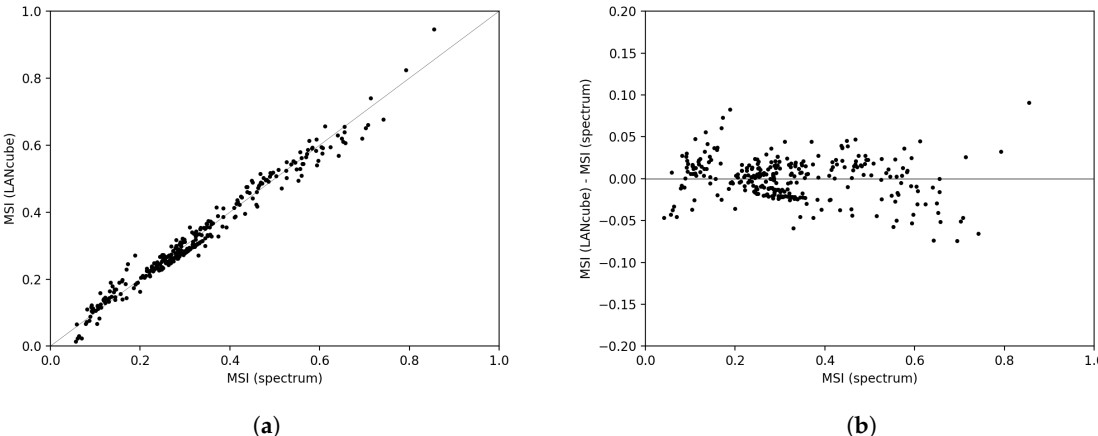

(**a**)                    (**b**)

**Figure 5.** Comparison of the LAN$^3$ derived MSI vs. MSI calculated with the spectrum. Panel (**a**) is showing the scatter plot and the 1:1 slope, while panel (**b**) give the residuals MSI(LANcube)-MSI(spectrum).

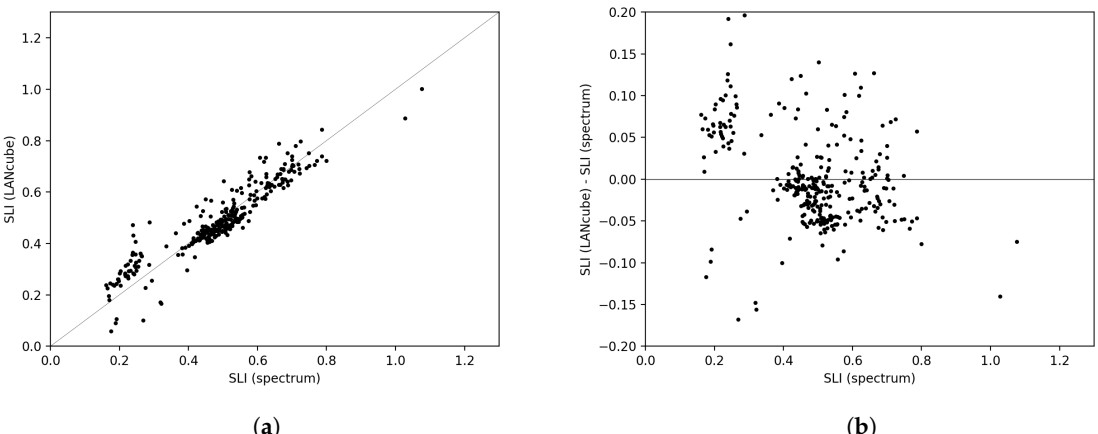

(**a**)                    (**b**)

**Figure 6.** Comparison of the LAN$^3$ derived SLI vs. SLI calculated with the spectrum. Panel (**a**) is showing the scatter plot and the 1:1 slope, while panel (**b**) give the residuals SLI(LANcube)-SLI(spectrum).

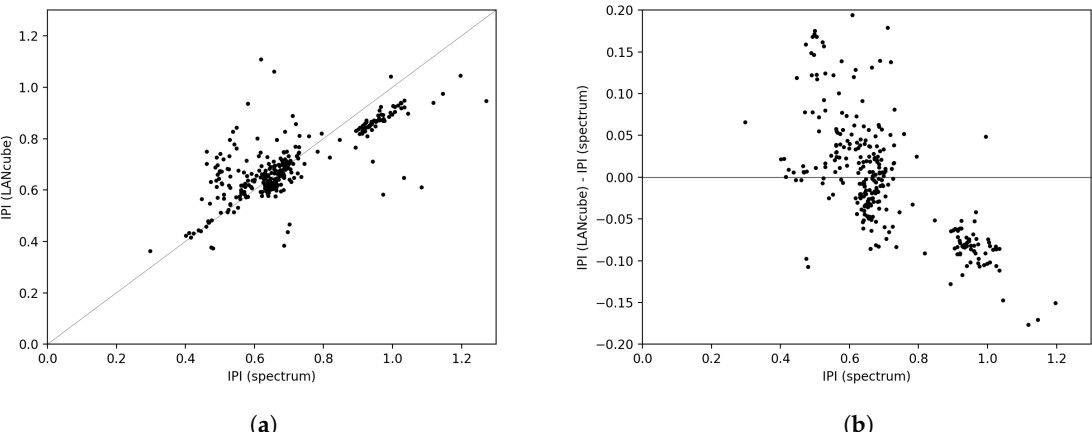

(**a**)                    (**b**)

**Figure 7.** Comparison of the LAN$^3$ derived IPI vs. IPI calculated with the spectrum. Panel (**a**) is showing the scatter plot and the 1:1 slope, while panel (**b**) give the residuals IPI(LANcube)-IPI(spectrum).

### 3.1. Maps of the MSI and SLI for Sherbrooke, Canada

As a direct result of the method that we developed, and to demonstrate the new possibility it delivers, we scanned the region of Sherbrooke City in Québec, Canada using the LAN[3] device installed on top of a car. Sherbrooke is a city of ∼160,000 inhabitants spread over an area of ∼350 km$^2$. Sherbrooke is the sixth largest city in the province of Québec and the thirtieth largest in Canada. It is located in the southern part of the Québec province, next to the US border. Sherbrooke is part of the first International DarkSky Reserve. Its commitment to protecting the night sky translates in a lighting regulation that restricts the blue light content of any new lighting installation. As a result, a significant part of the street lights are phosphor-converted amber LEDs.

During the experiment, the LAN[3] was aligned parallel to the moving direction and only the lateral sensors (toward houses) were used. Before the scan, we verified that the car headlights were not impacting the lateral sensors measurements by turning on and off the headlights. Using the lateral sensors allows us to form a better idea on the light pollution that can enter bedrooms and potentially impact human health. We have circulated in the city streets for 12 non-consecutive evenings. The $r$, $g$ and $b$ are calculated with data $R$, $G$, $B$ and $C$ data collected with the LAN[3] using Equations (5)–(7),

$$IR \approx \frac{R' + G' + B' - C'}{2} \tag{5}$$

$$r \approx R' - IR \qquad g \approx G' - IR \qquad b \approx B' - IR \tag{6}$$

with

$$R' = R/A/t_i \qquad G' = G/A/t_i \qquad B' = B/A/t_i \tag{7}$$

where $A$ and $t_i$ are, respectively, the sensor gain and the integration time(s) recorded by the LAN[3]. Note that in Equation (7), $t_i$ is expressed in seconds while the LAN[3] is recording it in units of milliseconds. The recorded values have to be divided by 1000 prior to using Equation (7). In general, it is better to make the correction because the spectral response curves of Figure 1a were determined for $gain = 1$ and $t_i = 1$ s. However, in the present study, it is not mandatory because, for a given sensor, the gain and integration time are the same for all bands and then taking the bands' ratios cancel their effect. In the LAN[3], the gain varies from 1 to 60 and the integration time from 2.4 to 614 ms depending on the ambient light level.

The $\left[\frac{b}{g}\right]$ and $\left[\frac{r}{g}\right]$ values were converted into MSI, SLI and IPI values using Equation (4). We excluded extrapolated data that are outside the span of values of data used for the fit. This was done by excluding indices lower than zero and higher than two. During the calculation of an index, all recorded $R$, $G$, $B$ and $C$ values are subject to a threshold value set at 20. Such a threshold value ensure that the Signal to Noise Ratio is higher than 20 (i.e., noise represents less than 5% of the signal). The two criteria excluded 410 measurements out of 14,894 raw data. It represents less than 3% of the database. In order to generate the maps from the localised measurements, we used the nearest neighbour interpolation with a maximum interpolation distance of 30 m. The resulting maps are shown in Figures 8–10. The relation between the lighting technologies and the map index classes are given in Tables 3–5. The complete dataset used to produce the maps is available on the project github repository [30].

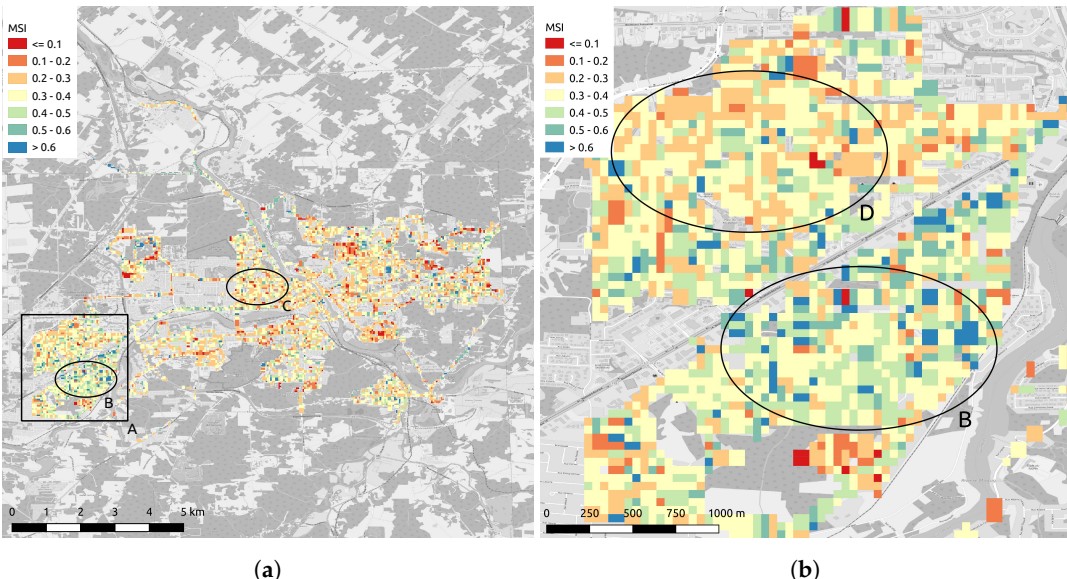

(**a**) (**b**)

**Figure 8.** Map of the MSI for Sherbrooke, Canada. We used Equation (4) and Table 2 with LAN[3] data. The panel (**b**) shows a zoomed view of MSI for the Mi-Vallon sector of Sherbrooke, Canada (black square A in panel (**a**)). Zones B–D are outlining three residential zones of Sherbrooke. In the map, blue colour is associated with the higher values.

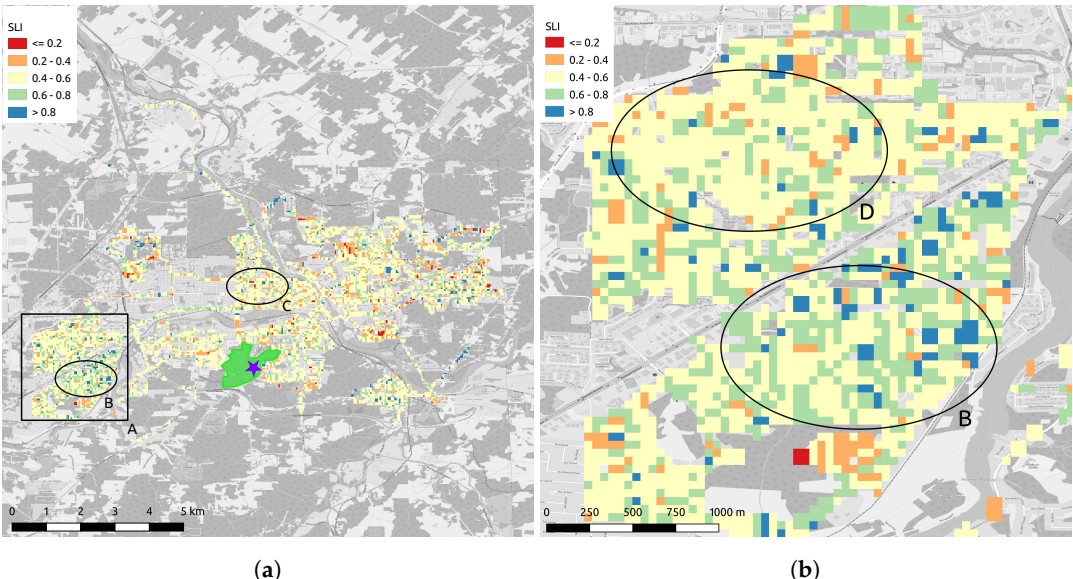

(**a**) (**b**)

**Figure 9.** Map of the SLI for Sherbrooke, Canada. We used Equation (4) and Table 2 with LAN[3] data. The panel (**b**) shows a zoomed view of SLI for the Mi-Vallon sector of Sherbrooke, Canada (black square A in panel (**a**)). Zones B–D are outlining three residential zones of Sherbrooke. The purple star is the epicentre of the Urban Dark Sky Oasis and the green polygon correspond to the limits of the protected area. In the map, blue colour is associated with the higher values.

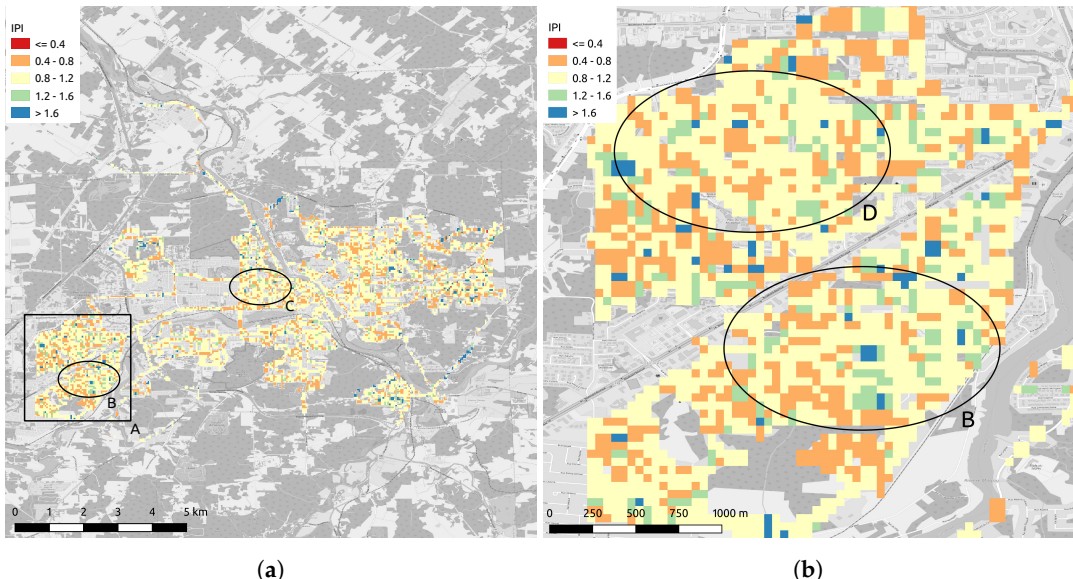

(**a**)                                                                                    (**b**)

**Figure 10.** Map of the IPI for Sherbrooke, Canada. We used Equation (4) and Table 2 with LAN[3] data. The panel (**b**) shows a zoomed view of IPI for the Mi-Vallon sector of Sherbrooke, Canada (black square A in panel (**a**)). Zones B–D are outlining three residential zones of Sherbrooke. In the map, blue colour is associated with the higher values.

**Table 3.** Lamps associated to mapped MSI ranges determined with the LSPDD. CFL stands for Compact Fluorescent, FL for fluorescent, WW for warm white, NW for neutral white and CW for cool white. In the map, blue colour is associated with the higher values.

| MSI Range | Lamp Spectra |
| --- | --- |
| $[0.0, 0.1)$ | LED PCamber, Yellow CFL, Red CFL |
| $[0.1, 0.2)$ | LED 2500 K, Incandescent, HPS |
| $[0.2, 0.3)$ | LED 2700 K, LED 3000 K, Incandescent, WW CFL |
| $[0.3, 0.4)$ | LED 3000 K, Incandescent, Halogene, WW CFL |
| $[0.4, 0.5)$ | LED 4000 K, NW CFL, NW FL |
| $[0.5, 0.6)$ | LED 5000 K, NW CFL, MH |
| $[0.6, 0.7)$ | LED 5000 K, NW CFL, CW CFL, MH |
| $[0.7, 0.8)$ | LED 6000 K, CW CFL |
| $[0.8, 0.9)$ | - |
| $[0.9, 1.0)$ | Blue CFL, Blue LED |

**Table 4.** Lamps associated to mapped SLI ranges determined with the LSPDD. CFL stands for Compact Fluorescent, FL for fluorescent, WW for warm white, NW for neutral white and CW for cool white.

| SLI Range | Lamp Spectra |
| --- | --- |
| $[0.0, 0.1)$ | Red CFL |
| $[0.1, 0.2)$ | PC amber LED |
| $[0.2, 0.3)$ | HPS, Yellow CFL |
| $[0.3, 0.4)$ | WW CFL, LED 2500 K |
| $[0.4, 0.5)$ | LED 2700 K, LED 3000 K, WW CFL Inc |
| $[0.5, 0.6)$ | LED 4000 K, Halogene, Incandescent |
| $[0.6, 0.7)$ | LED 4000 K, LED 5000 K, NW CFL, NW FL, Halogene, MH |
| $[0.7, 0.8)$ | LED 5000 K, LED 6000 K, CW CFL |
| $[0.8, 0.9)$ | CW CFL |
| $[0.9, 1.0)$ | Blue CFL, LED 6500 K |

**Table 5.** Lamps associated to mapped IPI ranges determined with the LSPDD. CFL stands for Compact Fluorescent, FL for fluorescent, WW for warm white, NW for neutral white and CW for cool white.

| IPI Range | Lamp Spectra |
|---|---|
| $[0.0, 0.1)$ | - |
| $[0.1, 0.2)$ | - |
| $[0.2, 0.3)$ | - |
| $[0.3, 0.4)$ | - |
| $[0.4, 0.5)$ | Yellow CFL |
| $[0.5, 0.6)$ | HPS, WW CFL, PC amber LED |
| $[0.6, 0.7)$ | WW CFL, NW CFL, LED 2700 K, LED 3000 K, LED 4000 K |
| $[0.7, 0.8)$ | NW CFL, CW CFL, LED 3000 K, LED 4000 K |
| $[0.8, 0.9)$ | Halogene |
| $[0.9, 1.0)$ | Halogene, Incandescent, Blue CFL, Red CFL |

## 4. Discussion and Conclusions

We estimate that the equations found to convert the $r$, $g$ and $b$ values of the LAN$^3$ into the three spectral indices—MSI, SLI and IPI—are suited for the mapping of large territories. The margin of error (95 percent confidence interval) is $\pm 0.05$ for the MSI, $\pm 0.1$ for the SLI and $\pm 0.2$ for the IPI. The margin of error for MSI appears to be sufficiently small for many health and ecosystems studies. For the SLI, the margin of error is larger but it is still useful as a tool for night sky protection. Tables 3 and 4 show that the margins of errors associated with the MSI and the SLI are small enough to separate the type of light technologies in regard to their spectral content. Therefore, we should expect the maps produced to be precise enough for most applications. The margin of error is much larger for the IPI and therefore we think this index requires further analysis in order to find better suited colour ratios to establish a better index calculation formula.

The index maps show that the averaged value all over the city for the MSI, SLI and IPI are 0.34, 0.54 and 0.96, respectively. Such numbers can be used to estimate a city performance to protect its territory against ALAN related issues. As an example, for the MSI, the average value is relatively small (0.34). As an element of comparison, this value is typical of 3000 K lamps. However, as the sensor detects all light emissions, this value is the result of a combination of both public and private lights. The sensitivity to private lights is probably higher given that we are using the lateral sensors and in Sherbrooke private lights are generally installed at a lower height. We used the lateral sensors because that it correspond to the orientation of propagation of the light that can enter the houses. In the city of Sherbrooke, the street lights are mainly composed of High-Pressure Sodium (HPS) (MSI $\approx 0.12$; [14]) and PC Amber LEDs (MSI $\approx 0.04$; [14]).

The indices maps allow us to identify critical zones in terms of their expected potential impact. As an example, Figure 8 shows only few values above MSI = 0.6 in the city of Sherbrooke (blue points in Figure 8a). Such data correspond to critical locations with respect to their potential impact on the melatonin suppression. In Sherbrooke, many of them are located in the main commercial zones. The residential Mi-Vallon sector is an exception (square A of Figure 8a,b for a zoomed-in view). The peculiarity of that residential area is clear when comparing zone B and zone C in Figure 8a. Zone C presents lower values of MSI but both zones are residential. In the Mi-Vallon sector, we can even distinguish two regimes between the northern (zone D in Figure 8b) and southern zones (zone B in Figure 8b). Zone B shows a much higher occurrence of high MSI values, but both zones are residential. They actually differ mainly by their date of construction, with zone D being the most recent. Lower MSI values in zone D actually reflect that the city administration is now gradually installing/converting street lights to PC amber, but another reason for this difference is that more houses are equipped with front door white bulbs in zone B than in zone D. Such private lamps pose a potential threat to citizens' health, but as they are under the control of the citizens themselves, it highlights the need for public outreach measures. Fortunately, we can find many places with very low MSI (<0.1) in the city. This situation happens because of the new city policy to replace or install

street lights by using PC amber LEDs and sometimes 2200K LEDs along with the absence of white private lamps.

A similar analysis can be made for the SLI and IPI index with Figures 9 and 10 respectively. One should keep in mind that the margin of error for SLI is twice that of MSI and that IPI margin of error is a 4-fold of the MSI. In the case of SLI, the blue pixel corresponds to SLI higher than 0.8. These places are the one to consider first when it is time to restore the starry sky. Such information is highly strategic in the case of Sherbrooke because of the current project under way to create an Urban Dark Sky Oasis. This project aims to give access to the Milky Way to the citizens in the Mont Bellevue city park. The limits of the park are identified by a green polygon in Figure 9a.

The mapping method presented here can rapidly identify the potentially harmful installations and then prepare targeted interventions to reduce or eliminate the problem. In a future work, we plan to find empirical relationships between the RGB and illuminance (lux) and the Correlated Colour Temperature (CCT). One other improvement could be to search for better set of colour ratios, including the clear channel, that could increase the correlation between the fitted equation and the MSI, the SLI and the IPI established with spectral data. This later task would be particularly useful for the IPI index given its large associated margin of error.

**Author Contributions:** We applied the sequence-determines-credit approach [31] for the sequence of authors which is in order of decreasing contributions. Conceptualisation, M.A. and J.R.; methodology, M.A. and C.M.; software, M.A. and C.M.; validation, M.A., C.M., A.D. and A.F.; formal analysis, M.A., C.M., A.F., A.D., A.S., J.Z. and C.T.; investigation, M.A., C.M., J.Z., C.T., A.F. and A.D.; resources, M.A.; data curation, C.M. and M.A.; writing—original draft preparation, M.A., A.F., A.D. and C.M.; writing—review and editing, M.A., C.M., A.F., A.D., J.R., J.Z., A.S. and C.T.; visualisation, C.M., M.A. and A.S.; supervision, M.A., J.R. and C.M.; project administration, M.A.; funding acquisition, M.A. and J.R. All authors have read and agreed to the published version of the manuscript.

**Funding:** M.A., A.F. and C.M. thank the Fonds de recherche du Québec–Nature et technologies (FRQNT) for financial support through the Research program for college researchers. J.R., M.A., A.F. and A.D. thank the Pôle régional en enseignement supérieur de l'Estrie (PRESE).

**Acknowledgments:** We want to thank Thomas Goulet-Soucy for his contribution to the design of the prototype version of the LAN³. We also want to thank Elsa Vigneau, Mathieu Fréchette and Thomas Goulet-Soucy for their help during the initial LAN³ testing and calibration experiments in Barcelona, Madrid, Lyon and Toulouse. Finally, we want to thank DH éclairage Inc. who built and graciously provided us the LAN³v1 systems used to map the city of Sherbrooke.

**Conflicts of Interest:** The authors declare no conflict of interest.

## Abbreviations

The following abbreviations are used in this manuscript.

| | |
|---|---|
| ADU | Analog to Digital Unit |
| ALAN | Artificial Light at Night |
| CFL | Compact fluorescent |
| CIE | Commission Internationale de l'Éclairage |
| DALAN | Direct Artificial Light at Night |
| DSLR | Digital single-lens reflex camera |
| FRQNT | Fonds de recherche du Québec–Nature et technologies |
| GPS | Global Positioning System |
| HPS | High-Pressure Sodium |
| IPI | Induced Photosynthesis Index |
| LED | Light-Emitting Diode |
| LSPDD | Lamp Spectral Power Distribution Database |
| MSAS | Melatonin Suppression Action Spectrum |
| MSI | Melatonin Suppression Index |

nm　　　　　nanometre
PAS　　　　Photosynthesis Action Spectrum
PC　　　　　Phosphor Converted
PRESE　　　Pôle régional en enseignement supérieur de l'Estrie
SLI　　　　　Star Light Index
UPS　　　　Uninterruptible Power Supply

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
