# Peer review of "Mapping the Melatonin Suppression, Star Light and Induced Photosynthesis Indices with the LANcube"

_remotesensing, doi:10.3390/rs12233954_

Round 1

Reviewer 1 Report

This is a very good manuscript explaining the use and capacities for ground monitoring with a multi-color device, so congratulations for the design of the instrument and the proposal of applications for ALAN studies. 

I think the current version of the document is almost ready but in my opinion there are some actions that have to be done and some recommendations for its improvement. I will go section by section explaining my recommendations and also identifying some minor errors.

* Abstract and Introduction

Both sections are well described and introducing the topic and the concept of this manuscript. But I propose some improvement to connect what is said in the abstract. In current lines 6 to 9 it is exposed the existence of space instruments and ground techniques to evaluate ALAN but later this is not described or discussed on the rest of manuscript in comparison with the new device. 

So maybe in the final paragraph of the introduction section it could fit perfectly to introduce some lines to comment space measurements and ground measurements with a couple of general references of their application to ALAN measurements (for example Falchi et al 2016 for satellite measurements applied to World Atlas of ALAN or Hanel et al 2018 for the global perspective of preexisting ground techniques and devices ). With this you can justity why LanCube is the powerful tool for your aims in front of all other devices.

Also on Introduction on line 40 there is the reference of Sun spectrum, the use of a citation there seems needed. If just D65 is the only possible source of this information, maybe it should appear on reference/bibliography format and not using parenthesis.

A minor spelling error appears on line 54. “The SLI. T also..”. “. T” seems a typo. 

* Materials and methods

On section 2.1 I found some problems with the general text and its connection with figures that I will summarize below:

- The first paragraph is a bit confusing for reader, it is not enough clear in comparison with the good writing of the rest of the manuscript. So maybe authors can re-read and evaluate an improvement for clear reading.

- Also in this first paragraph on current line 85 it sounds me strange the use of “from the outside world” to refer to environment.

- The main problem with this section is the strategy of the figures. Now reading the text the first one to be cited is figure 1a and then 2a-2b. So figure 1b is not cited until next subsection. This means the reader will go up and down looking for figures. So my proposal is to put as figure 1 current number 2 (the one of the devices) and introduce it for example on current line 82 “..six faces.” So after that current figure 1 (now would be figure 2) would be cited in order with text.

- Around current figure 1. There is a small problem with panel (a), the caption (a) appears partially blocked by plot of panel (b). In the caption of this figure it is necessary to add “In panel (a) we show..” in the beginning to identify that first part of the caption refers to panel (a). 

On section 2.2 there are small problems of writing (typos or similar):

- On current line 113 a “. T,” appears that it seems a typo.

- On current line 124 the name of spectrometer is “Stellarnet” now appears “Sellarnet”

- The sentence started on current line 130 and ending in 131 seems a “lost sentence”. The information is said very similar above (current line 121) and ends with an uncomplete citation to equations 5 and 6. These equations appear many pages below and are not connected with this section.

* Results

This section is in general well focused and presents the results in a good way, so I will just add some comments for a possible improvement of the section.

- On current line 155 when writing around the fitting of Figure 4 appears “is quite good”. I think that this is just a very subjective comment and it is quite easy to provide some fitting quality parameters for any kind of fitting to give a real idea of the quality. I know that later some tests are showed but this sentences here is a bit inaccurate.

- Around table 3 is just a subsample of the total lamps dataset used. Maybe it could be better to create a supplementary dataset in a repository with the whole content and skip this table with subsample of the total set.

* Discussions and Conclusions

Around this section I think it is well focused too and I do not have any specific comment on it.

Author Response

Reviewer 1:

This is a very good manuscript explaining the use and capacities for ground monitoring with a multi-color device, so congratulations for the design of the instrument and the proposal of applications for ALAN studies.

I think the current version of the document is almost ready but in my opinion there are some actions that have to be done and some recommendations for its improvement. I will go section by section explaining my recommendations and also identifying some minor errors.

R: Thank you for the constructive comments.

* Abstract and Introduction

Both sections are well described and introducing the topic and the concept of this manuscript. But I propose some improvement to connect what is said in the abstract. In current lines 6 to 9 it is exposed the existence of space instruments and ground techniques to evaluate ALAN but later this is not described or discussed on the rest of manuscript in comparison with the new device.

So maybe in the final paragraph of the introduction section it could fit perfectly to introduce some lines to comment space measurements and ground measurements with a couple of general references of their application to ALAN measurements (for example Falchi et al 2016 for satellite measurements applied to World Atlas of ALAN or Hanel et al 2018 for the global perspective of preexisting ground techniques and devices ). With this you can justity why LanCube is the powerful tool for your aims in front of all other devices.

R: Very good suggestion an additional paragraph have been added to the introduction.

Also on Introduction on line 40 there is the reference of Sun spectrum, the use of a citation there seems needed. If just D65 is the only possible source of this information, maybe it should appear on reference/bibliography format and not using parenthesis.

R: Corrected – added reference

A minor spelling error appears on line 54. “The SLI. T also..”. “. T” seems a typo.

R: Thanks

* Materials and methods

On section 2.1 I found some problems with the general text and its connection with figures that I will summarize below:

- The first paragraph is a bit confusing for reader, it is not enough clear in comparison with the good writing of the rest of the manuscript. So maybe authors can re-read and evaluate an improvement for clear reading.

R: The paragraph have been rewritten in order to make it clearer.

- Also in this first paragraph on current line 85 it sounds me strange the use of “from the outside world” to refer to environment.

R: Removed

- The main problem with this section is the strategy of the figures. Now reading the text the first one to be cited is figure 1a and then 2a-2b. So figure 1b is not cited until next subsection. This means the reader will go up and down looking for figures. So my proposal is to put as figure 1 current number 2 (the one of the devices) and introduce it for example on current line 82 “..six faces.” So after that current figure 1 (now would be figure 2) would be cited in order with text.

R: We think that the problem is now resolved.

- Around current figure 1. There is a small problem with panel (a), the caption (a) appears partially blocked by plot of panel (b). In the caption of this figure it is necessary to add “In panel (a) we show..” in the beginning to identify that first part of the caption refers to panel (a).

R: Corrected

On section 2.2 there are small problems of writing (typos or similar):

- On current line 113 a “. T,” appears that it seems a typo.

- On current line 124 the name of spectrometer is “Stellarnet” now appears “Sellarnet”

- The sentence started on current line 130 and ending in 131 seems a “lost sentence”. The information is said very similar above (current line 121) and ends with an uncomplete citation to equations 5 and 6. These equations appear many pages below and are not connected with this section.

R: Corrected.

* Results

This section is in general well focused and presents the results in a good way, so I will just add some comments for a possible improvement of the section.

- On current line 155 when writing around the fitting of Figure 4 appears “is quite good”. I think that this is just a very subjective comment and it is quite easy to provide some fitting quality parameters for any kind of fitting to give a real idea of the quality. I know that later some tests are showed but this sentences here is a bit inaccurate.

R: Thank for this really good comment. We removed that subjective sentence.

- Around table 3 is just a subsample of the total lamps dataset used. Maybe it could be better to create a supplementary dataset in a repository with the whole content and skip this table with subsample of the total set.

R: We removed the table and refered to the complete data as a link to the github repository.

* Discussions and Conclusions

Around this section I think it is well focused too and I do not have any specific comment on it.

Reviewer 2 Report

Overall wee written and sound. Just a few minor issues:

lines 45-46, Even 3000K LED used in household and outdoor lighting have in percentage more blue light than sunshine, jet authors quite that MSI is seldom larger than 1 in typical lightnig techs. I suggest to clarify this with a few sentences.

The sensor IR sensitivity (Fig. 1) seems quite high. Does it have an IR block filter, and if it has, why is it so inefficient. I feel that blocking the IC will be more sound method than making corrections described in the article.

Figures 5 and 6: the dots are very small and their colors are hard to discern. Please, enlarge them to make colors more distinguishable.

Figure 9 or appropriate place in text: I am quite proud of my general geography knowledge, yet I do not know anything about Sherbrooke. I suppose most non-canadian readers will feel the same. I suggest adding a short description of the town, i.e. general location, population, evt. some remarks on lighting policy there.

Author Response

Reviewer 2

Overall wee written and sound. Just a few minor issues:

R: Thank you very much

lines 45-46, Even 3000K LED used in household and outdoor lighting have in percentage more blue light than sunshine, jet authors quite that MSI is seldom larger than 1 in typical lightnig techs. I suggest to clarify this with a few sentences.

R: This disagreement with our sentence rely probably on the definition we have of the blue. For that reason we remove that reference to the blue percentage.

The sensor IR sensitivity (Fig. 1) seems quite high. Does it have an IR block filter, and if it has, why is it so inefficient. I feel that blocking the IC will be more sound method than making corrections described in the article.

R: Yes, this is an interesting avenue for future version of the LANcube. According to the manufacturer there is a IR blocking filter but obviously it is not very efficient.

Figures 5 and 6: the dots are very small and their colors are hard to discern. Please, enlarge them to make colors more distinguishable.

R: We kept the size the same but remove the distinction between in situ and lab measurements. We estimate that this information was not really mandatory. The small points allow to see the amount of data used. We think the new figures will be more clear.

Figure 9 or appropriate place in text: I am quite proud of my general geography knowledge, yet I do not know anything about Sherbrooke. I suppose most non-canadian readers will feel the same. I suggest adding a short description of the town, i.e. general location, population, evt. some remarks on lighting policy there.

R: This paragraph was added

Sherbrooke is a city of ~160,000 inhabitants spreaded over an area of $~350 km^2. Sherbrooke is the sixth largest city in the province of Québec and the thirtieth largest in Canada. It is located in the southern part of the Québec province, next to the US border. Sherbrooke is part of the first International DarkSky Reserve. Its commitment to protect the night sky translates in a lighting regulation that restricts the blue light content of any new lighting installation. As a result, a significant part of the street lights are phosphor converted amber LEDs.

Reviewer 3 Report

Manuscript ID:  remotesensing-1015225

Journal:  Remote Sensing

Title:  Mapping the Melatonin Suppression, Star Light and Induced Photosynthesis Indices with the LANcube

Summary:  The manuscript describes how a newly developed device, the LANcube, can be used to map three indices that reflect the potential impacts of artificial light at night (ALAN): the melatonin suppression index, the star light index, and the induced photosynthesis index.  Such mapping effort could be used to identify areas within cities that are especially impacted by ALAN, and target mitigation efforts.  

My expertise is not in remote sensing, but rather with the biological impacts of ALAN.  Thus, I may not be fully qualified to critique the methodology of this paper.  Considering this, I do not have any major criticisms of the paper, which I believe has the potential to make a nice contribution to the field.  I have provided some minor comments that may help improve the presentation.

L20.  Do you mean both diurnal and nocturnal animals?  The sentence structure makes this somewhat unclear.

L24.  And also circadian rhythms in other animals.

L31.  Provide a citation for monitoring the level of melatonin in biofluids.

L33-62.  Are the values of the three different indices correlated with each other?  In other words, to what extent is the information conveyed by these three indices redundant versus distinct?

L70.  Perhaps best to mention here, and also perhaps in the Abstract, that the LANcube is a system that you have newly developed.

L103.  Clarify what this limitation is.  The length of the acquisition time?

Fig 5-7.  The green triangles are somewhat hard to see.  Perhaps you could use a different color.

L157.  Provide a brief explanation regarding why this is the most important color ratio.

L172-175.  Some questions about the sampling protocol:  Were the car lights on or off?  How would this impact the readings? Between what hours was the survey run, and was it all on one night?  Was the car constantly in motion?

L186.  How much extrapolated data was it necessary to exclude?

Fig 8.  In this figure, higher values are associated with blue, which I guess is because light in the blue spectra has higher MSI.  However, my brain at first associates the red with higher values (since this is often done for other types of data).  So, maybe it would be good to highlight in the caption that blue is associated with higher values for this reason.

L207.  Issues related to ALAN exposure?

L210-211.  Why were the other sensors also not used?

Language suggestions

L37.  Should be “typically ranges”

L105.  “have” should be “has”

L153-154.  change to: “can capture the various points of the dataset relatively well”.

L155.  Remove “actually”

L165.  Which reflect a standard deviation sounds strange, and also the double use of more.  Rephrase.

L181.  Remove “that”

L214.  Should be “zones”

L222.  Should be “shows”

L225.  Change “such” to “this”

L236.  Change “undergoing” to “underway”

Author Response

Reviewer 3

The manuscript describes how a newly developed device, the LANcube, can be used to map three indices that reflect the potential impacts of artificial light at night (ALAN): the melatonin suppression index, the star light index, and the induced photosynthesis index. Such mapping effort could be used to identify areas within cities that are especially impacted by ALAN, and target mitigation efforts.

My expertise is not in remote sensing, but rather with the biological impacts of ALAN. Thus, I may not be fully qualified to critique the methodology of this paper. Considering this, I do not have any major criticisms of the paper, which I believe has the potential to make a nice contribution to the field. I have provided some minor comments that may help improve the presentation.

R: Many thanks for your valuable expertise input. The LANcube was designed to help to study the biological impact of ALAN, your review is of great interest to us.

L20. Do you mean both diurnal and nocturnal animals? The sentence structure makes this somewhat unclear.

R: corrected

L24. And also circadian rhythms in other animals.

R: Corrected

L31. Provide a citation for monitoring the level of melatonin in biofluids.

R: added

L33-62. Are the values of the three different indices correlated with each other? In other words, to what extent is the information conveyed by these three indices redundant versus distinct?

R: Yes the MSI and the SLI are somewhat correlated the relationship between them being relatively linear but with an offset. The relation ship show an important dispersion for high MSI values. IPI is not well correlated to the two others because that its spectral sensitivity quite different.

L70. Perhaps best to mention here, and also perhaps in the Abstract, that the LANcube is a system that you have newly developed.

R: Done. Thanks

L103. Clarify what this limitation is. The length of the acquisition time?

R: Actually yes, but in response to another reviewer we changed this part of the paragraph.

Fig 5-7. The green triangles are somewhat hard to see. Perhaps you could use a different color.

R: Very good comment. We decided to use only the black points for all the data. We estimate that it is not useful to the study to differentiate the origin of the spectra.

L157. Provide a brief explanation regarding why this is the most important color ratio.

R: We added a sentence to explain it with the higher valuer of the beta and epsilon coefficient.

L172-175. Some questions about the sampling protocol: Were the car lights on or off? How would this impact the readings? Between what hours was the survey run, and was it all on one night? Was the car constantly in motion?

R: Because that we are only considering the lateral sensors, the effect of the headlights is very small. We tested it before scanning the city by turning the lights on and off. We added a sentence about that. The scan was executed over many evenings.

L186. How much extrapolated data was it necessary to exclude?

R: Excellent point. Both the extrapolated data excluded and the criteria about SNR>20 removed 410 point out of 14894. This information is now added to the article.

Fig 8. In this figure, higher values are associated with blue, which I guess is because light in the blue spectra has higher MSI. However, my brain at first associates the red with higher values (since this is often done for other types of data). So, maybe it would be good to highlight in the caption that blue is associated with higher values for this reason.

R: Ok

L207. Issues related to ALAN exposure?

R: Done

L210-211. Why were the other sensors also not used?

R: added to the text : We used the lateral sensors because that it correspond to the orientation of propagation of the light that can enter the houses.

Language suggestions

L37. Should be “typically ranges”

L105. “have” should be “has”

L153-154. change to: “can capture the various points of the dataset relatively well”.

L155. Remove “actually”

L165. Which reflect a standard deviation sounds strange, and also the double use of more. Rephrase.

L181. Remove “that”

L214. Should be “zones”

L222. Should be “shows”

L225. Change “such” to “this”

L236. Change “undergoing” to “underway”

R: Done. Many thanks
